# Proteomic Characterization of the Olfactory Molecular Imbalance in Dementia with Lewy Bodies

**DOI:** 10.3390/ijms21176371

**Published:** 2020-09-02

**Authors:** Mercedes Lachén-Montes, Naroa Mendizuri, Domitille Schvartz, Joaquín Fernández-Irigoyen, Jean Charles Sánchez, Enrique Santamaría

**Affiliations:** 1Clinical Neuroproteomics Unit, Navarrabiomed, Complejo Hospitalario de Navarra (CHN), Universidad Pública de Navarra (UPNA), Irunlarrea 3, 31008 Pamplona, Spain; mercedes.lachen.montes@navarra.es (M.L.-M.); naroa.mendizuri.sanchez@navarra.es (N.M.); jfernani@navarra.es (J.F.-I.); 2Proteored-ISCIII. Proteomics Platform, Navarrabiomed, Complejo Hospitalario de Navarra (CHN), Universidad Pública de Navarra (UPNA), Irunlarrea 3, 31008 Pamplona, Spain; 3IdiSNA, Navarra Institute for Health Research, Spain Irunlarrea 3, 31008 Pamplona, Spain; 4Translational Biomarker Group, Department of Medicine, University of Geneva, Rue Michel Servet 1, 1211 Geneve, Switzerland; Domitille.Schvartz@unige.ch (D.S.); Jean-Charles.Sanchez@unige.ch (J.C.S.)

**Keywords:** olfaction, proteomics, olfactory bulb, dementia, Lewy bodies

## Abstract

Olfactory dysfunction is one of the prodromal symptoms in dementia with Lewy bodies (DLB). However, the molecular pathogenesis associated with decreased smell function remains largely undeciphered. We generated quantitative proteome maps to detect molecular alterations in olfactory bulbs (OB) derived from DLB subjects compared to neurologically intact controls. A total of 3214 olfactory proteins were quantified, and 99 proteins showed significant alterations in DLB cases. Protein interaction networks disrupted in DLB indicated an imbalance in translation and the synaptic vesicle cycle. These alterations were accompanied by alterations in AKT/MAPK/SEK1/p38 MAPK signaling pathways that showed a distinct expression profile across the OB–olfactory tract (OT) axis. Taken together, our data partially reflect the missing links in the biochemical understanding of olfactory dysfunction in DLB.

## 1. Introduction

Dementia with Lewy bodies (DLB) constitutes one of the most common causes of dementia only behind Alzheimer’s disease (AD), accounting for approximately 25% of dementia cases [1,2]. Clinically, DLB is manifested by fluctuating cognition with severe variation in attention, recurrent formed and detailed visual hallucinations, and spontaneous parkinsonism motor features. Other common characteristics are temporary consciousness loss, delusions, syncope, repeated falls, systematized hallucinations and neuroleptic sensitivity [2,3,4,5]. In general, all these symptoms are preceded by rapid eye movement sleep disorder, psychiatric symptoms, dysautonomia together with occipital hypo-metabolism, cognitive impairment and smell loss [6,7]. Olfactory impairment has been demonstrated in a plethora of neurological disorders, including AD, Parkinson’s disease (PD) and DLB [8,9]. Although not currently implemented in the clinic, olfactory assessment has been postulated as a promising tool for the differential diagnosis of AD and DLB [10,11,12,13,14]. Previous clinicopathological studies have shown that olfactory dysfunction is associated with Lewy body pathology in limbic and neocortical regions [15], and to date, most of the studies have revealed that olfactory deficits are more severe in DLB than in AD [8,13,14,16]. However, despite the great attention that has caught in the last decade, the sense of smell is often undermine in the clinical diagnosis, and clinicians rarely test this deficit in patients with suspicion of neurological disease.

DLB is pathologically characterized by the presence of Lewy bodies and Lewy neurites in the brainstem, limbic system, and cortical areas [17]. Those are insoluble aggregates of abnormal α-synuclein, which is phosphorylated, nitrated and truncated, producing oligomeric species that afterwards constitute the so-called Lewy bodies and Lewy neurites [18,19,20,21]. Thus, together with Parkinson’s disease (PD) and others, DLB is classified among α-synucleopathies. Importantly, although some controversy still exists, DLB is differentiated from PD dementia (PDD) in the cognitive impairment onset. Following the international guidelines, DLB is diagnosed when cognitive impairment precedes parkinsonian motor symptoms or begins within one year from its onset [22]. On the other hand, cognitive impairments develop after a well-established parkinsonism in PDD patients [23].

Lewy pathology has also been found in the olfactory system, including the olfactory bulb (OB) and the anterior olfactory nucleus (AON) [24,25]. Recent reports have demonstrated the presence of α-synuclein pathology in the OB in 97% of DLB cases [24]. Interestingly, the Braak staging model has suggested that α-synuclein pathology starts at the OB and the AON, progressing afterwards in a caudorostral direction through vulnerable interconnected neurons [26]. This so-called prion hypothesis suggests that the formation of α-synuclein does not occur in a cell-autonomous mechanism. Instead, this process occurs in a determined number of cells and then spreads to the neighboring neuronal cells, reaching distant brain regions [27,28]. Interestingly, this has been demonstrated by the α-synuclein inoculation in the OB and its posterior spreading through among 23 different brain regions [29]. Nevertheless, whether this cell-to-cell spreading mechanisms is the main responsible for the neurodegenerative process remains unknown. In addition, information regarding the OB molecular alterations occurring in consequence is still lacking.

The immense cellular heterogeneity that comprises the central nervous system still represents a big challenge for the study of brain-related disorders. Nowadays, high-throughput techniques such as proteomics have emerged as a valuable approach to deep into the complex molecular mechanisms that underlie the process of neurodegeneration [30,31,32]. Thus, tissue-based quantitative clinical proteomics has already been aimed to provide more insights into DLB pathology [33,34,35,36]. However, although olfactory proteomics has been successfully applied in neurodegenerative phenotypes in mice and humans [37,38,39,40,41,42,43,44,45,46], the olfactory proteostatic imbalance that occurs in DLB remains unknown. To our knowledge, this is the first study aiming to decipher the specific molecular mechanisms that underlie the process of neurodegeneration at the level of primary olfactory areas in DLB.

## 2. Results and Discussion

Due to the fact that DLB progression is accompanied by an acute olfactory dysfunction [8] and α-synuclein propagation from the OB has been proposed as a potential onset of this pathology [29], we have performed an in-depth quantitative proteomic study in combination with functional interaction data and biochemical approaches to determine the dysregulation of the OB proteome in DLB (Figure 1A). Post-mortem OB specimens were subjected to isobaric chemical tags (tandem mass tags, TMT) coupled to mass spectrometry (four cases/condition). Among 3214 quantified proteins (Appendix A), differential analysis revealed 99 deregulated proteins in OBs derived from DLB subjects compared to neurological intact controls (16 up-regulated and 83 downregulated proteins) (Table 1). It is important to note that due to the technical workflow used, we failed to accurately quantify many proteins expressed at low levels that might also participate in the DLB phenotype. Moreover, our study does not distinguish between the numerous cell types coexisting in the OB, so in principle, it is not possible to directly assign protein changes to a specific cell type. To partially overcome these drawbacks, and based on prior quantitative thresholding of RNA-seq abundance data, a cell type enrichment analysis across OB quantified proteome and the OB differential dataset was performed using previous characterized cell type-enriched marker genes derived from different OB cell populations [47]. As shown in Figure 1B, cluster-enriched genes corresponding to different neuronal subtypes are represented in the quantified OB proteome. Interestingly, part of the proteostatic alterations are specifically enriched to granular cells (PRUNE2, STXBP1, CAMK2A, PSD3, PENK, RPS27, CAMK2B, GRIA2, TUBB2A, PSIP1, PRKAR1B, SYNPR, NAPB), astrocytes (RPL37, RPL31), external plexiform layer (EPL) interneurons (SCAMP5), mitral/Tufted cells (RAB3B, SYNGR3, ATP1A3, SLC17A7, MYCBP2, NRXN1, SYP, ATP1B1, CD47, GRM1, REEP5, NCEH1, ATP6V0A1, SV2A, CNTNAP2), olfactory sensory neurons (OSN) (RPL22, RPL36, RPL23A, RPS25, RPL38, RPS2, RPL23), and periglomerular cells (LY6H, SLC32A1, RAB3A, ATP6V1A) (Figure 1C). All these data shed new light on the molecular disturbances that accompany the neurodegenerative process across each cellular homeostasis in DLB. With the aim to complement and validate quantitative proteome measurements, subsequent experiments were performed to check the steady-state levels of a subset of differential proteins using downstream assays. As shown in Figure 1D, PI16 and ALDH1B1 are up-regulated in DLB at the level of the OB, confirming the mass spectrometry data. While this is the first report linking PI16 with any neurodegenerative process, mitochondrial ALDH1B1 plays a crucial role concerning the oxidative metabolism of toxic aldehydes in the brain. Thus, the alteration of ALDH1B1 in the OB together with the recent emerging evidence for ALDH2 stands for its implication in the process of neurodegeneration [48].

### 2.1. Olfactory Protein Translation and Synaptic Function are Impaired in DLB

To characterize in detail the olfactory proteotype in DLB phenotypes, we performed proteome-scale interaction networks merging the 99 differential proteins detected at the level of the OB. As shown in Figure 2A, the protein interactome was mainly composed by specific protein clusters related to translation, unblocking of NMDA receptors, synaptic vesicle cycle and innate immune system. Principally, OB proteostatic imbalance affects translation-related processes, defined by the significant downregulation of a great number of ribosomal proteins (Figure 2A). In this context, alteration in mRNA expression of ribosomal proteins has been previously observed in frontal cortex from DLB patients [49], suggesting a common pathological mechanism in these two brain regions. Moreover, alterations in protein synthesis have also been related to the presence of α-synuclein oligomers in PD-related brain areas [50]. Accompanying the reduction in EEF1A2 levels, the disruption of OB protein synthesis was confirmed by the drop in the P70 S6K beta (ribosomal protein S6 kinase beta) (Figure 2B), a serine/threonine kinase involved in the selective translation of unique family of mRNAs coding the ribosomal proteins and elongation factors [51]. In accordance with our data, reduced protein levels of the highly homologous EEF1A1 isoform have been observed in the synaptic fraction of DLB brain specimens [52], suggesting a dysregulation of the synaptic protein translation in DLB. Accordingly, several types of dementia, including DLB, are characterized by an important loss of synaptic transmission [53,54]. Additional functional analyses have commonly pointed out potential alterations in olfactory synapsis-related processes (Figure 2A,C). These data are in agreement with alterations found at prefrontal cortical level across DLB and PDD pathologies, where a significant loss of multiple synaptic proteins has been reported [33]. In addition, among those synapsis-related deregulated proteins, Rab3b significantly downregulated in the OB of DLB subjects has been recently proposed as a predictive marker for cognitive decline in DLB patients [33].

### 2.2. Specific Olfactory Derangements in Survival Pathways in DLB

We performed additional proteome-scale interactive networks using the deregulated OB proteome. Using the Ingenuity Pathway Analysis (IPA) (QIAGEN Redwood City, www.qiagen.com/ingenuity), both AKT (protein kinase B) and ERK (extracellular signal-regulated kinase 1/2) appeared as main nodes in protein interactome maps (Figure 3A). Although we did not detected changes in the expression of these survival kinases in the proteomic approach, alterations in some of their targets may be in accordance with their potential alteration at the level of the OB in DLB cases. In fact, an activation of AKT is observed in DLB cases, in parallel with an increment in the phosphorylation of BAD (Bcl2-associated agonist of cell death) at Ser112 (Figure 3B). This serine is a canonical target of AKT, modulating apoptosis [55]. This phosphorylation phenomenon has been related with BAD binding to 14-3-3t protein, which sequesters BAD from BclxL [56]. Thus, the inhibition of the proapoptotic factor pBAD, together with the tendency to increase in BclxL protein suggests readjustments in the survival potential of the OB from DLB subjects. Previous reports have demonstrated a differential deregulation of multiple survival kinases in AD and PD at the level of the OB [39,40,42]. Subsequent experiments were performed to monitor upstream and downstream effectors of the following specific routes: (i) MAPK pathway: ERK1/2 (extracellular signal-regulated kinase 1/2) and MEK1/2 (mitogen-activated protein kinase kinase 1/2), (ii) SEK1 (mitogen-activated protein kinase kinase 4)-SAPK/JNK (stress-activated protein kinase/Jun-amino terminal kinase) axis, and (iii) the MKK3/6 (dual specificity mitogen-activated protein kinase kinase 3/6)-p38 MAPK axis. Although still under controversy, AKT and ERK have been previously related to α-synuclein deposits [57,58]. However, the expression profiling detected in DLB (Figure 3C) differs from the observed during AD or PD progression. No alterations in AKT signaling have been previously observed during both AD and PD stage-dependent OB analysis [39,42]. On the other hand, previous work has demonstrated the activation of olfactory ERK survival factor during AD [42]. Interestingly, although both PD and DLB are characterized by the presence of α-synuclein deposits, an opposite pattern concerning ERK protein levels was observed in PD at the level of the OB [39]. Moreover, while no alterations are found in upstream MEK in DLB OBs (Appendix A), a significant increase was observed in advanced stages of PD [39]. Altogether, these results point out that olfactory AKT/MAPK pathways are differential molecular features between both alpha-synucleinopathies. Moreover, stress-responsive kinases targeted by activated AKT [59] were evaluated in DLB OBs. Only a clear increase in OB SEK1 levels was noticed in DLB (Figure 3C), while no appreciable changes were detected in its activated status (data not shown). On the contrary, an activation of olfactory SEK1 has been observed in AD subjects [42]. Otherwise, downstream targets such as SAPK/JNK (stress-activated protein kinase/Jun-amino terminal kinase) or PKAc (protein kinase A) were unchanged (Appendix A). Although the p38 MAPK cascade has been previously related to PD pathogenesis [60], no specific information is available concerning its involvement in DLB. The p38 MAPK signaling route mediates numerous cell effects in response to external signals and interestingly, an early activation has been observed in AD OBs [42]. However, no such alterations have been observed for PD [39]. In our analysis, a significantly increment in p38 MAPK levels was observed in OB specimens derived from DLB cases (Figure 3C). These alterations were not observed in two of its main upstream kinases, MKK3 and MKK6 (dual specificity mitogen-activated protein kinase kinase 3/6) (Appendix A).

### 2.3. Survival Kinome Differs across the OB–OT Axis in DLB

The olfactory tract (OT) is constituted by the axons coming from the mitral and tufted neurons located in the OB. A significant degeneration of axons has been detected in the OT from AD subjects [61]. In pre-AD mild cognitive impairment subjects, the loss of fiber OT integrity corresponds to a loss of gray matter density in parallel with a reduced glucose metabolism in central olfactory structures [62,63]. Moreover, OTs undergo early and sequential morphological alterations that correlate with the development of dementia [64]. On the other hand, atrophy and changes in the structural integrity of OT have been also observed in PD subjects compared to controls [65]. The deposition of pathological α-synuclein protein has been observed not only in the OB but also in the olfactory tract (OT) [9]. Considering the fact that the analysis of the OT area may provide clues regarding the molecular alterations along the olfactory system during the neurodegenerative process, subsequent experiments were performed to characterize the survival kinome in this region. Our analysis revealed the following: (a) OT-specific alterations in MEK, AKT and PKA (Figure 4); (b) opposite alterations between OB and OT protein levels for SEK1 and p38 MAPK (Figure 4); and (c) unchanged OB/OT protein levels for SAPK1 and MKK3/MKK6 (Appendix A). Altogether, these results point out different protein expression profiles across the OB–OT axis, laying the foundation for future exhaustive molecular profiling of OT areas in order to increase our understanding about the smell deficits that accompany the neurodegenerative process in alpha-synucleinopathies.

## 3. Materials and Methods

### 3.1. Materials

The following reagents and materials were used: anti-ALDH1B1 (ref. sc-374090) was purchased from Santa Cruz; anti-PI16 (ref. PAS 31882) from Thermo; anti-P70 S6K beta (ref: ab184551) from Abcam; and anti-Bcl-xL (ref. 2764), anti-pAkt (Ser473) (ref. 4060), anti-Akt (ref. 4685), anti-pMEK1/2 (Ser217/221) (ref. 9154), anti-MEK1/2 (ref. 9126), anti-pERK1/2 (Thr202/Tyr204) (ref. 4370), anti-ERK1/2 (ref. 9102), anti-pPKA (Thr197) (ref. 5661), anti-PKA (ref. 4782), anti-pSEK1/MKK4 (Ser257/Thr261) (ref. 9156), anti-SEK1/MKK4 (ref. 9152), anti pSAPK/JNK (Thr183/Tyr185) (ref. 9255), anti pSAPK/JNK (ref. 9252S), anti pMKK3-6 (Thr183/Tyr185) (ref. 9231), anti MKK3 (ref. 5674), anti p-p38 MAPK (Thr180/Tyr182) (ref. 4511), anti p38 MAPK (ref. 9212) were purchased from Cell Signaling. Electrophoresis reagents were purchased from Biorad and trypsin from Promega.

### 3.2. Human Samples

According to the Spanish Law 14/2007 of Biomedical Research, written consent forms of the Neurological Tissue Bank of Navarra Health Service and Neurological Tissue Bank of IDIBAPS-Hospital Clinic (Barcelona, Spain) were obtained for research purposes. All assessments, post-mortem evaluations, and procedures were approved by the Clinical Ethics Committee of Navarra (code 2016/36) and were conducted in accordance with the Declaration of Helsinki. For the proteomic phase, four DLB cases were analyzed, while four cases from elderly subjects with no history or histological findings of any neurological disease were used as controls. All cases had a post-mortem interval (PMI) lower than 16 h. For the validation phase, OB tissue from additional subjects was included (*n* = 12) (Table 2). Neuropathological evaluation was performed according to standardized guidelines [2].

### 3.3. Sample Preparation for Proteomic Analysis

OB specimens derived from DLB cases and non-neurological subjects were homogenized using “mini potters’’ in lysis buffer containing 8 M urea and 0.1 M TEAB as previously described [39]. The homogenates were spun down at 14,000× *g* for 15 min at 4 °C. Protein quantitation was performed with the Bradford assay kit (Bio-Rad, Hercules, CA, USA).

### 3.4. Reduction, Alkylation, Digestion and TMT Labeling

For the proteomic analysis, 25 µg of protein from each sample was used. Proteins were first reduced with 10 mM TCEP for 1 h at 30 °C. Then, alkylation was performed for 30 min at room temperature using 40 mM IAA. Samples were diluted 5 times to reduce the urea concentration <1 M. Trypsinization and peptide labeling was performed as previously described [39]. Control cases were labeled with 127N, 127C, 128N, and 128C and disease cases were labeled with 129N, 129C, 130N and 130C (Labels 126 and 131 were used as intern quality control).

### 3.5. Off-Gel Electrophoresis and LC-MS/MS Analysis

Samples previously purified with C18 Macro Spin Columns (Harvard Apparatus) were fractionated using off-gel electrophoresis (OGE) separation using an Agilent 3100 off-gel fractionator as previously described [66]. Samples were then desalted and purified using MicroSpins C18 columns, dried in a speed-vacuum and stored at −20 °C until analysis. An LTQ Orbitrap Q-exactive Plus mass spectrometer (Thermo Fisher, Waltham, MA, USA) coupled with nano-HPLC was used to analyzed the OGE fractions [42]. Peptides were reconstituted using 5% ACN and 0.1% FA; were trapped on a 2 cm × 75 µm ID, 3 µm pre-column; and separated on a 50 cm × 75 µm ID, PepMap C18, 2 µm easy-spray column (Thermo Scientific, Waltham, MA, USA). The analytical separation was run for 60 min using a gradient of H2O/FA (99.9%/0.1%; solvent A) and CH3CN/FA (99.9%/0.1%; solvent B) at a flow rate of 300 nL/min. For MS survey scans, the OT resolution, the ion population, the number of precursor ions and the collision energy used have been previously described [39].

### 3.6. Protein Identification and Quantification

MS data were processed using EasyProt. Peak list was obtained using OGE fractions, and the combination of HCD-CID raw data peak list was generated. These data were submitted to the EasyProt software platform (version 2.3, build 718) that uses Phenyx software (GeneBio, Geneva, Switzerland) for protein identification [67]. The search was performed using the Uniprot/Swiss-Prot database (2014-10, 66903) using Homo Sapiens taxonomy; oxidized methionine (variable modification); and TMT-10 amino-terminus, TMT-10 lysine and cysteine carbamethylation (fixed modifications). One missed cleavage was selected, and parent-ion tolerance was set to 10 ppm and the accuracy of fragment ions to 0.6 Da. Only proteins with a false discovery rate (FDR) lower than 1% and at least 2 unique peptides were considered (peptide length of 6 amino acids). Isobaric quantification was performed using the Isobar R package [68]. To correct the isotopic impurities of TMT-10 reporter ion intensities, the manufacturer’s isotopic distribution data were applied. To normalize the reporter intensities, the equal median intensity method was considered. Biological and technical variability were calculated for each protein ratio in order to test the ratio’s accuracy and biological significance. For each variable, a ratio and sample *p*-values were calculated. Proteins with a cut-off threshold value higher than 1.33 or lower than 0.77 were considered. All MS data files have been deposited to the ProteomeXchange Consortium [69] (http://proteomecentral.proteomexchange.org) via the PRIDE partner repository with the dataset identifier PXD016069 (username: reviewer98884@ebi.ac.uk; password: O8KNli2b).

### 3.7. Bioinformatics

The proteomic information was analyzed using STRING (Search Tool for the Retrieval of Interacting Genes) software (v.11) (http://stringdb.org/) [70] to infer protein interactomes and detect differentially activated/deactivated pathways as a result of DLB phenotypes. The identification of specifically dysregulated regulatory/metabolic networks in DLB was analyzed through the use of QIAGEN’s Ingenuity Pathway Analysis (QIAGEN Redwood City, www.qiagen.com/ingenuity).

### 3.8. Western-Blotting

Equal amounts of protein (10 µg) were resolved in 4–15% stain free SDS-PAGE gels (Bio-rad). OB/OT proteins derived from control and DLB subjects were electrophoretically transferred onto nitrocellulose membranes as previously described [39]. Membranes were probed with primary antibodies at 1:1000 dilution in 5% nonfat milk or BSA and using a horseradish peroxidase-conjugated secondary antibody at a 1:5000 dilution. Enhanced chemiluminescence (Perkin Elmer, Madrid, Spain) was used to visualize the immunoreactivity using a Chemidoc MP Imaging System (Bio-Rad). Equal loading in the gels was evaluated using stain-free imaging technology [71]. Thus, protein normalization was performed by measuring the total protein directly on the gels used for Western blotting. Image Lab Software (v.5.2; Bio-Rad) was used for densitometric analyses. Optical density values were normalized to total protein levels and considered as arbitrary units.

## 4. Conclusions

Overall, the present study provides new insights regarding the molecular mechanisms governing the olfactory dysfunction in DLB. Besides the pathological alpha-synuclein depositions previously observed in olfactory areas, we have demonstrated a disarrangement in the OB proteostasis, affecting translation and the synaptic vesicle cycle and showing disturbances in the protein expression profile of cell survival routes across the OB–OT axis. Thus, our findings provide basic information for understanding the implication of olfactory structures in the pathophysiology of DLB, identifying protein mediators that may be used as potential therapeutic agents or even explored in biofluids as candidate biomarkers for DLB diagnosis and evolution.

## Figures and Tables

**Figure 1 ijms-21-06371-f001:**
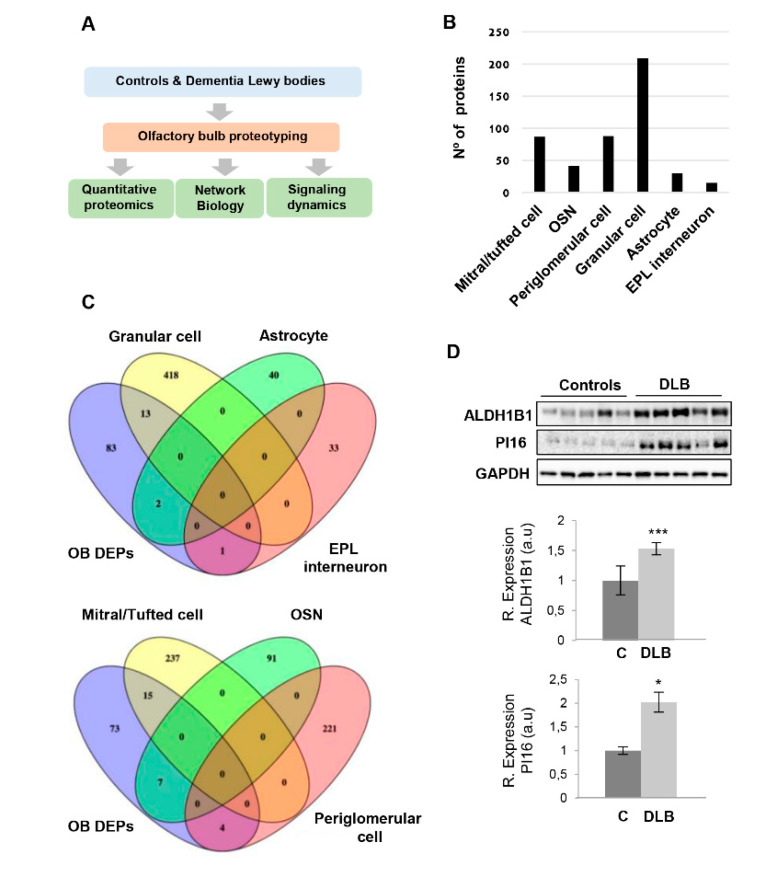
(**A**) An overview of the workflow used for the molecular characterization of the olfactory bulbs (OB) derived from dementia with Lewy bodies (DLB) subjects. (**B**) Quantified proteome distribution across OB cell layers. (**C**) Cluster-enriched genes in specific OB cell layers that are differentially expressed at the level of the OB in DLB subjects. (**D**) OB protein expression changes of ALDH1B1 and PI16 in DLB subjects by Western blotting. Data are presented as mean ± SEM. * *p* < 0.05 vs. control group; *** *p* < 0.001 vs. control group (a.u: arbitrary units; DEPs: differential expressed proteins; EPL: external plexiform layer; OSN: olfactory sensory neuron).

**Figure 2 ijms-21-06371-f002:**
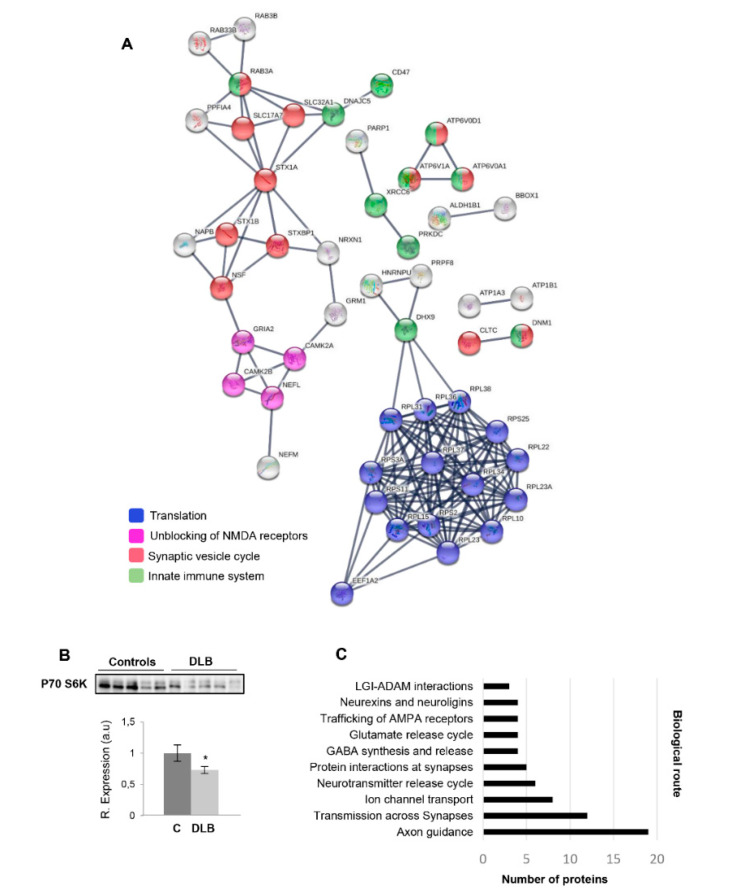
Impairment of OB protein translation and synaptic function in DLB. (**A**) Interactome network for OB deregulated proteome using STRING (Search Tool for the Retrieval of Interacting Genes). Proteins are represented with nodes and the physical/functional interactions with continuous lines. Interactions tagged as “high confidence” (>0.7) in the STRING database were exclusively considered. K means clustering was used. (**B**) Western-blotting of P70 S6K protein levels across controls and DLB subjects. Data are presented as mean ± SEM. * *p* < 0.05 vs. control group. In this case, equal loading of the gel was assessed using stain-free imaging technology, and protein normalization was performed by measuring total protein directly on the gels (a.u; arbitrary units). (**C**) Significantly enriched synapsis-related pathways in the OB differential proteome using the STRING tool.

**Figure 3 ijms-21-06371-f003:**
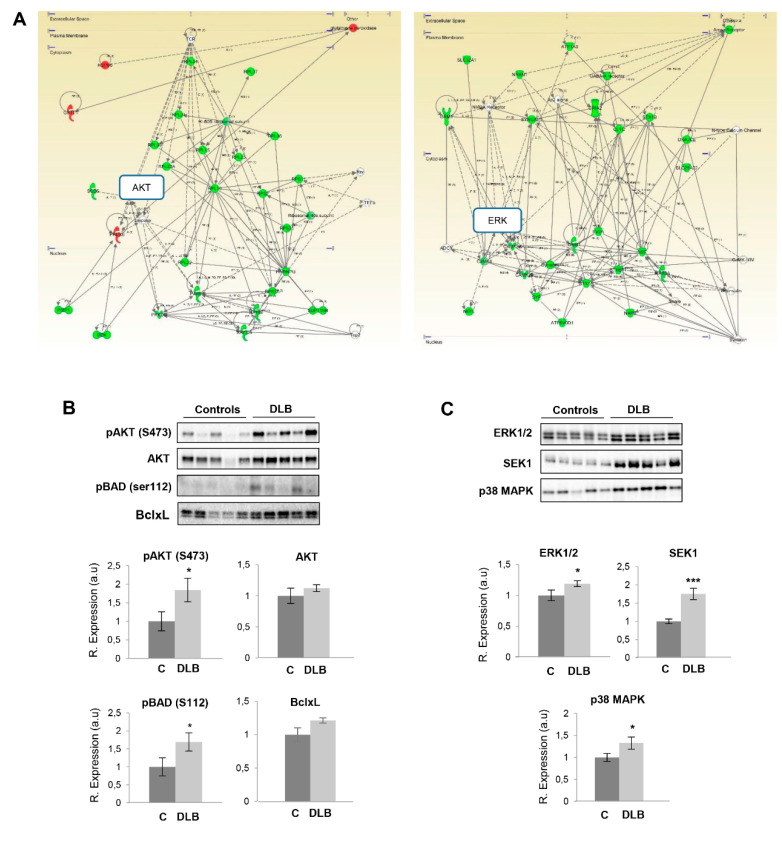
Disruption of OB signaling routes in DLB. (**A**) Visual representation of protein interactome maps for OB differentially expressed proteins in DLB. Upregulated proteins in red and downregulated proteins in green. Continuous lines represent direct interactions, while discontinuous lines correspond to indirect functional interactions. See complete legend and high resolution images in Appendix A. (**B**) Activation state and protein measurements of AKT and Bcl2-associated agonist of cell death (BAD) across controls and DLB subjects by Western-blotting. (**C**) OB ERK1/2, SEK1 and p38 MAPK levels at the level of OB. Data are presented as mean ± SEM. * *p* < 0.05 vs. control group; *** *p* < 0.001 vs. control group. Equal loading of the gels was assessed using stain-free imaging technology, and protein normalization was performed by measuring total protein directly on the gels (a.u; arbitrary units).

**Figure 4 ijms-21-06371-f004:**
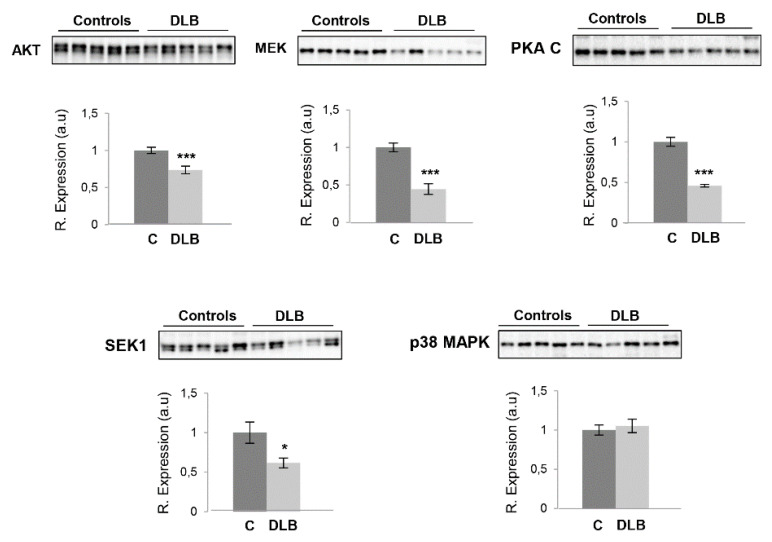
Disruption of olfactory tract (OT) signaling routes in DLB. Steady-state levels of AKT, MEK, PKAc, SEK1 and p38 MAPK in the OTs derived from controls and DLB subjects. Data are presented as mean ± SEM. * *p* < 0.05 vs. control group; *** *p* < 0.001 vs. control group. Equal loading of the gels was assessed using stain-free imaging technology, and protein normalization was performed by measuring total protein directly on the gels (a.u; arbitrary units).

**Table 1 ijms-21-06371-t001:** Significantly deregulated proteins in DLB OB specimens.

Description	Gene Name	Uniprot	Peptide Count	Ratio	*p* Value Sample
**Down-regulated proteins across DLB**					
60S ribosomal protein L34	RPL34	P49207	2	0.44	4.6181 × 10^−13^
60S ribosomal protein L37	RPL37	P61927	2	0.465	7.3563 × 10^−14^
Protein prune homolog 2	PRUNE2	Q8WUY3	3	0.5454	4.0993 × 10^−10^
60S ribosomal protein L22	RPL22	P35268	2	0.574	1.4869 × 10^−12^
Ras-related protein Rab-3B	RAB3B	P20337	2	0.6125	6.995 × 10^−9^
60S ribosomal protein L36	RPL36	Q9Y3U8	2	0.6139	2.0299 × 10^−10^
V-type proton ATPase subunit d 1	ATP6V0D1	P61421	10	0.6144	6.1715 × 10^−11^
Heterogeneous nuclear ribonucleoprotein U	HNRNPU	Q00839	20	0.6181	1.9382 × 10^−11^
DNA-dependent protein kinase catalytic subunit	PRKDC	P78527	19	0.6264	1.5042 × 10^−11^
FACT complex subunit SPT16	SUPT16H	Q9Y5B9	2	0.6328	8.6739 × 10^−11^
Synaptogyrin-3	SYNGR3	O43761	3	0.6506	5.5741 × 10^−11^
Liprin-alpha-4	PPFIA4	O75335	1	0.654	4.3343 × 10^−10^
60S ribosomal protein L10	RPL10	P27635	2	0.6672	2.1869 × 10^−10^
60S ribosomal protein L23a	RPL23A	P62750	3	0.6675	4.0283 × 10^−10^
40S ribosomal protein S11	RPS11	P62280	3	0.6679	4.1716 × 10^−9^
HMG nucleosome × 10−binding domain-containing protein 3	HMGN3	Q15651	2	0.6692	1.4543 × 10^−9^
Sodium/potassium-transporting ATPase subunit alpha-3	ATP1A3	P13637	26	0.6693	5.6496 × 10^−11^
Lymphocyte antigen 6H	LY6H	O94772	5	0.6734	2.4723 × 10^−9^
40S ribosomal protein S25	RPS25	P62851	3	0.6741	1.1199 × 10^−9^
Neurofilament light polypeptide	NEFL	P07196	18	0.6743	7.1158 × 10^−10^
40S ribosomal protein S3a	RPS3A	P61247	4	0.6782	3.4891 × 10^−9^
Vesicular inhibitory amino acid transporter	SLC32A1	Q9H598	6	0.6816	2.3798 × 10^−10^
Vesicular glutamate transporter 1	SLC17A7	Q9P2U7	2	0.6818	2.7575 × 10^−11^
Plasma membrane calcium-transporting ATPase 3	ATP2B3	Q16720	2	0.686	4.3062 × 10^−9^
E3 ubiquitin-protein ligase MYCBP2	MYCBP2	O75592	2	0.6862	3.7771 × 10^−10^
Syntaxin-1A	STX1A	Q16623	7	0.6879	4.8927 × 10^−9^
Syntaxin-binding protein 1	STXBP1	P61764	29	0.6904	1.4785 × 10^−10^
Neurofilament medium polypeptide	NEFM	P07197	27	0.6932	1.5816 × 10^−9^
Secretory carrier-associated membrane protein 5	SCAMP5	Q8TAC9	4	0.6946	2.3559 × 10^−10^
Calcium/calmodulin-dependent protein kinase type II subunit alpha	CAMK2A	Q9UQM7	8	0.7012	5.1862 × 10^−11^
Neurexin-1	NRXN1	Q9ULB1	1	0.7014	7.3383 × 10^−10^
PH and SEC7 domain-containing protein 3	PSD3	Q9NYI0	11	0.7037	1.2708 × 10^−9^
Gamma-aminobutyric acid receptor subunit beta-2	GABRB2	P47870	2	0.7042	1.5422 × 10^−9^
Proenkephalin-A cleaved into synenkephalin; Met-enkephalin	PENK	P01210	5	0.7134	3.9046 × 10^−8^
Synaptophysin	SYP	P08247	5	0.7156	5.4513 × 10^−11^
40S ribosomal protein S27, 40S ribosomal protein S27-like	RPS27	Q71UM5	2	0.7165	1.5132 × 10^−9^
Syntaxin-1B	STX1B	P61266	13	0.7165	3.4018 × 10^−10^
DnaJ homolog subfamily C member 5	DNAJC5	Q9H3Z4	5	0.7172	2.3736 × 10^−10^
Calcium/calmodulin-dependent protein kinase type II subunit beta	CAMK2B	Q13554	11	0.7184	3.952 × 10^−10^
Mitochondrial glutamate carrier 1	SLC25A22	Q9H936	2	0.7204	9.6273 × 10^−7^
Sodium/potassium-transporting ATPase subunit beta-1	ATP1B1	P05026	13	0.7216	7.7815 × 10^−10^
Glutamate receptor 2	GRIA2	P42262	5	0.7238	2.3865 × 10^−9^
Tubulin beta-2A chain	TUBB2A	Q13885	2	0.7246	1.4984 × 10^−9^
PC4 and SFRS1-interacting protein	PSIP1	O75475	11	0.7258	1.7917 × 10^−9^
60S ribosomal protein L38	RPL38	P63173	2	0.7272	3.6241 × 10^−9^
Clathrin heavy chain 1	CLTC	Q00610	58	0.7305	1.1969 × 10^−9^
Ras-related protein Rab-3A	RAB3A	P20336	4	0.7312	2.1299 × 10^−9^
Leukocyte surface antigen CD47	CD47	Q08722	3	0.7319	3.2189 × 10^−8^
Metabotropic glutamate receptor 1	GRM1	Q13255	3	0.7365	4.3436 × 10^−9^
Receptor expression-enhancing protein 5	REEP5	Q00765	5	0.7394	8.049 × 10^−9^
Probable leucine--tRNA ligase, mitochondrial	LARS2	Q15031	2	0.7395	7.1964 × 10^−9^
Neutral cholesterol ester hydrolase 1	NCEH1	Q6PIU2	4	0.7407	1.4583 × 10^−9^
40S ribosomal protein S2	RPS2	P15880	7	0.7411	1.147 × 10^−8^
Glucose 1,6-bisphosphate synthase	PGM2L1	Q6PCE3	8	0.7438	3.5479 × 10^−9^
60S ribosomal protein L15	RPL15	P61313	3	0.7446	4.3186 × 10^−8^
Transmembrane protein 35A	TMEM35A	Q53FP2	2	0.7446	3.4577 × 10^−9^
60S ribosomal protein L23	RPL23	P62829	2	0.7448	1.9037 × 10^−7^
Disintegrin and metalloproteinase domain-containing protein 22	ADAM22	Q9P0K1	2	0.746	2.1643 × 10^−8^
60S ribosomal protein L31	RPL31	P62899	3	0.7469	2.7904 × 10^−9^
Pre-mRNA-processing-splicing factor 8	PRPF8	Q6P2Q9	5	0.7484	3.9116 × 10^−10^
V-type proton ATPase 116 kDa subunit a isoform 1	ATP6V0A1	Q93050	16	0.7514	2.8591 × 10^−9^
Actin-related protein 3B	ACTR3B	Q9P1U1	5	0.752	6.086 × 10^−9^
ATP-dependent RNA helicase A	DHX9	Q08211	20	0.7521	6.8201 × 10^−9^
cAMP-dependent protein kinase type I-beta regulatory subunit	PRKAR1B	P31321	1	0.7522	2.2148 × 10^−7^
Ras/Rap GTPase-activating protein SynGAP	SYNGAP1	Q96PV0	7	0.7527	2.5633 × 10^−8^
Stearoyl-CoA desaturase 5	SCD5	Q86SK9	2	0.7552	1.0027 × 10^−6^
Elongation factor 1-alpha 2	EEF1A2	Q05639	7	0.7557	3.9797 × 10^−9^
Synaptic vesicle glycoprotein 2A	SV2A	Q7L0J3	6	0.7558	3.2828 × 10^−9^
Protein rogdi homolog	ROGDI	Q9GZN7	3	0.7569	5.4595 × 10^−9^
X-ray repair cross-complementing protein 6	XRCC6	P12956	15	0.7573	2.6216 × 10^−10^
Protein DEK	DEK	P35659	2	0.7598	1.2282 × 10^−8^
Vesicle-fusing ATPase	NSF	P46459	32	0.7608	2.2491 × 10^−9^
Cytochrome c oxidase subunit 7A-related protein, mitochondrial	COX7A2L	O14548	2	0.7613	2.1107 × 10^−8^
Ras-related protein Rab-33B	RAB33B	Q9H082	1	0.7614	5.1009 × 10^−7^
Kelch-like protein 22	KLHL22	Q53GT1	2	0.7633	2.2287 × 10^−8^
Dynamin-1	DNM1	Q05193	22	0.7641	1.0564 × 10^−8^
Probable G-protein coupled receptor 158	GPR158	Q5T848	2	0.7653	1.1577 × 10^−6^
Poly [ADP-ribose] polymerase 1	PARP1	P09874	10	0.7654	4.795 × 10^−7^
Synaptoporin	SYNPR	Q8TBG9	2	0.7657	5.4958 × 10^−9^
Beta-soluble NSF attachment protein	NAPB	Q9H115	14	0.7666	2.9345 × 10^−8^
V-type proton ATPase catalytic subunit A	ATP6V1A	P38606	23	0.7674	1.3052 × 10^−8^
Contactin-associated protein-like 2	CNTNAP2	Q9UHC6	7	0.7674	1.2516 × 10^−8^
ATPase inhibitor, mitochondrial	ATPIF1	Q9UII2	6	0.769	5.9459 × 10^−8^
***Up-regulated proteins across DLB***					
Glutathione S-transferase theta-1	GSTT1	P30711	4	1.34	6.8584 × 10^−8^
Phosphoglycerate mutase 2	PGAM2	P15259	4	1.3405	2.2404 × 10^−9^
Cysteine and glycine-rich protein 1	CSRP1	P21291	12	1.3412	5.701 × 10^−9^
Phosphotriesterase-related protein	PTER	Q96BW5	4	1.3447	2.9204 × 10^−10^
Peroxiredoxin-6	PRDX6	P30041	15	1.357	2.3469 × 10^−9^
Phenazine biosynthesis-like domain-containing protein	PBLD	P30039	2	1.3623	8.8245 × 10^−6^
Gamma-butyrobetaine dioxygenase	BBOX1	O75936	5	1.4003	8.8403 × 10^−10^
Heat shock protein beta-6	HSPB6	O14558	2	1.4074	4.9675 × 10^−10^
GMP reductase 1	GMPR	P36959	4	1.4169	1.2597 × 10^−11^
Pirin	PIR	O00625	3	1.4254	3.3049 × 10^−9^
Protein S100-A4	S100A4	P26447	3	1.4504	2.1294 × 10^−9^
Aldehyde dehydrogenase X, mitochondrial	ALDH1B1	P30837	5	1.5159	1.3337 × 10^−10^
Nicotinate-nucleotide pyrophosphorylase [carboxylating]	QPRT	Q15274	4	1.516	5.6656 × 10^−8^
Metallothionein-1E	MT1E	P04732	1	1.5881	1.6841 × 10^−10^
Collagen alpha-3(VI) chain	COL6A3	P12111	28	1.9992	1.5501 × 10^−14^
Peptidase inhibitor 16	PI16	Q6UXB8	3	2.0955	1.2432 × 10^−10^

Down-regulated and up-regulated proteins in DLB OB are indicated in green and red respectively.

**Table 2 ijms-21-06371-t002:** Clinicopathological data of subjects included in this study.

Groups	Case (Code)	Age (Years)	Sex	PMI	OB	OT	Definitive DX	Proteomic Phase	Western Blot Analysis
Control	BK-0300	75	F	-	Yes	Yes	ARP I-II	x	x
	BK-1378	78	M	6 h	Yes	Yes	multi-infarct	x	x
	BK-1078	84	F	6 h	Yes	Yes	Vascular encephalopathy	x	x
	BK-1195	82	F	8 h	Yes	Yes	Acute stroke left cerebral artery	x	x
	BK-1563	79	M	5 h	Yes	Yes	Acute stroke left cerebral artery		x
DLB	CS-0622	78	M	8 h 30 min	Yes	Yes	DLB-AD neocortical	x	x
	CS-0870	74	M	15 h	Yes	Yes	DLB-AD neocortical	x	x
	CS-0938	80	M	9 h 40 min	Yes	Yes	DLB-AD neocortical	x	x
	CS-0947	72	M	11 h	Yes	Yes	DLB-AD neocortical	x	x
	CS-1096	73	F	15 h 30 min	Yes	Yes	DLB-AD neocortical		x
	CS-1140	82	F	4 h 30 min	Yes	Yes	DLB-AD neocortical		x
	CS-1158	74	M	10 h	Yes	Yes	DLB-AD neocortical		x
	CS-1215	67	F	13 h 30 min	Yes	Yes	DLB-AD neocortical		x
	CS-1282	78	F	5 h	Yes	Yes	DLB-AD neocortical		x
	CS-1192	77	F	14 h 20 min	Yes	Yes	DLB-AD neocortical		x
	432	71	F	9 h	Yes	No	DLB-AD neocortical		x
	397	90	F	5 h	Yes	No	DLB-AD neocortical		x
	394	70	M	4 h	Yes	No	DLB-AD neocortical		x
	339	70	M	4 h 15 min	Yes	No	DLB-AD neocortical		x
	325	91	M	12 h	Yes	No	DLB-AD neocortical		x

PMI: post-mortem interval; DX: diagnosis. “x” indicates that the sample has been used in the proteomic phase, Western-blot analysis or in both approaches.

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
