# Peer review of "Proteomic Characterization of the Olfactory Molecular Imbalance in Dementia with Lewy Bodies"

_ijms, 2020, doi:10.3390/ijms21176371_

Round 1
Reviewer 1 Report
in my opinion it is a well done study with somewhat confused but very interesting premises and, unfortunately, with the limit of having chosen to analyze cytological material taken from cadaver, in vivo actually it would have found higher concentrations of the proteins studied, presumably, and it would have been easier to amplify the genetic material. I would have liked some more insight into the possible implications of such an analysis in vivo and in clinical practice, the time it would take to do such an analysis to actually arrive at the diagnosis and an attempt, on their part, to define a cut-off to the above which we can define the molecular alteration in the olfactory tract there is no way to correlate clinical olfactory alteration with the molecular changes in cadavers.
Clear and intuitive tables.
Lines 31-32 à I mean, it was okay, I get what they meant, just poorly written
Lines 34-35 à As much as we’re concerned, it’s still not true in the clinical practice. Also, the premise is that we’re dealing with a non specific symptom which unfortunately occurs in a VERY LARGE spectrum of neurodegenerative disorders and, in general, in neurological diseases. It’s reasonable to expect an olfactory impairment in that group of patients, but how would you get to a differential diagnosis? Not clear.
39 – 40 à clear, but poorly written
83- 84 à So we don’t get the chance to know where those proteins come from in the first place? But we do afterwards? Why would you point that out? ...not clear!
128 – 129 à again, this proves that you can’t get to a differential diagnosis using olfactory impairment only!
Reviewer 2 Report
Lachen-Montes and colleagues have shown a proteomic profile of the olfactory molecular alterations in OB from DLB patients.
Overall, the paper is potentially interesting highliting the potential liks between biochemical changes and olfactory dysfunctions in DLB. However, the entire manuscript needs further work and improvement and important part are missing.
Major points
- The authors writes that the densitometry of the Western blots is a relative expression of the normalized data, but then in the axis is written arbitrary units which means that the measurements are absolute?
- The results part 2.2 starts analyzing signaling pathways; however, it is completely missing why the authors look at those pathways and what the single proteins are. An expert on the subject, of course, knows that but anyone else not.
- Why is the manuscript organized in Results and Discussion together? In fact, a real discussion and implementation with the literature up to date is lacking.
Author Response
Dear Editor,
Please find enclosed the revised version of the manuscript with reference IJMS-898328 entitled “Proteomic Characterization of the Olfactory Molecular Imbalance in Dementia with Lewy Bodies”. We wish to thank the editor and reviewers for their comments and suggestions that greatly improved the quality of this manuscript, and hope that the new version is now acceptable for its publication in International Journal of Molecular Sciences (Special issue: Molecular Research on Neurodegenerative Diseases 2.0).
All changes and modifications are highlighted in yellow throughout the manuscript. Specific paragraphs have been modified or introduced in the introduction section, results & discussion (section 2.2, section 2.3), material and methods and conclusions. Abbreviations list has been updated.
These references have been incorporated in the new version of the manuscript:
- IG McKeith, D Galasko, K Kosaka et al. Consensus guidelines for the clinical and pathologic diagnosis of dementia with Lewy bodies (DLB): report of the consortium on DLB international workshop. Neurology 1996 Nov;47(5):1113-24. doi: 10.1212/wnl.47.5.1113.
-Ian McKeith, Jacobo Mintzer, Dag Aarsland et al. Dementia with Lewy bodies. Lancet Neurol 2004 Jan;3(1):19-28. doi: 10.1016/s1474-4422(03)00619-7.
- IG McKeith, D W Dickson, J Lowe, M Emre et al. Diagnosis and management of dementia with Lewy bodies: third report of the DLB Consortium. Neurology 2005 Dec 27;65(12):1863-72. doi: 10.1212/01.wnl.0000187889.17253.b1. Epub 2005 Oct 19.
- Mary Catherine Mayo, Yvette Bordelon. Dementia with Lewy bodies. Semin Neurol. 2014 Apr;34(2):182-8. doi: 10.1055/s-0034-1381741. Epub 2014 Jun 25.
- Hiroshige Fujishiro, Shinichiro Nakamura, Kiyoshi Sato, Eizo Iseki. Prodromal dementia with Lewy bodies. Geriatr Gerontol Int 2015 Jul;15(7):817-26. doi: 10.1111/ggi.12466. Epub 2015 Feb 17.
- P C Donaghy, J T O'Brien, A J Thomas. Prodromal dementia with Lewy bodies. Psychol Med 2015 Jan;45(2):259-68. doi: 10.1017/S0033291714000816. Epub 2014 Apr 3.
- Thomas G Beach, Charles H Adler, Nan Zhang et al. Severe hyposmia distinguishes neuropathologically confirmed dementia with Lewy bodies from Alzheimer's disease dementia. PLoS One 2020 Apr 22;15(4):e0231720. doi: 10.1371/journal.pone.0231720. eCollection 2020.
- D C Davies, J W Brooks, D A Lewis. Axonal loss from the olfactory tracts in Alzheimer's disease
Neurobiol Aging Jul-Aug 1993;14(4):353-7. doi: 10.1016/0197-4580(93)90121-q.
- Philipp A Thomann, Vasco Dos Santos, Ulrich Seidl, Pablo Toro et al. MRI-derived atrophy of the olfactory bulb and tract in mild cognitive impairment and Alzheimer's disease. J Alzheimers Dis 2009;17(1):213-21. doi: 10.3233/JAD-2009-1036.
- Donna J Cross, Yoshimi Anzai, Eric C Petrie et al. Loss of olfactory tract integrity affects cortical metabolism in the brain and olfactory regions in aging and mild cognitive impairment. J Nucl Med
2013 Aug;54(8):1278-84. doi: 10.2967/jnumed.112.116558. Epub 2013 Jun 26.
-Praveen Bathini, Antoine Mottas, Muriel Jaquet, Emanuele Brai, Lavinia Alberi. Progressive signaling changes in the olfactory nerve of patients with Alzheimer's disease. Neurobiol Aging 2019 Apr;76:80-95. doi: 10.1016/j.neurobiolaging.2018.12.006. Epub 2018 Dec 27.
- Christoph Scherfler, Michael F Schocke et al. Voxel-wise analysis of diffusion weighted imaging reveals disruption of the olfactory tract in Parkinson's disease. Brain 2006 Feb;129(Pt 2):538-42. doi: 10.1093/brain/awh674. Epub 2005 Nov 4.
Answers to the reviewers:
REVIEWER 2
Lachen-Montes and colleagues have shown a proteomic profile of the olfactory molecular alterations in OB from DLB patients. Overall, the paper is potentially interesting highliting the potential liks between biochemical changes and olfactory dysfunctions in DLB. However, the entire manuscript needs further work and improvement and important part are missing.
The authors writes that the densitometry of the Western blots is a relative expression of the normalized data, but then in the axis is written arbitrary units which means that the measurements are absolute?
Thank you for the reviewer´s observation. First, we obtain the optic density value for each protein. Then, each intensity is normalized to total protein levels in each lane (based on the stain-free imaging technology mentioned in methods section-3.8). The data showed in the graphs represent the normalized intensity (in arbitrary units) not absolute values. To clarify this point, this sentence:
“ Image Lab Software (v.5.2; Bio-Rad) was used for densitometric analyses, and optical density values were considered as arbitrary units and normalized to total protein levels” has been changed by (lines 317-318):
“Image Lab Software (v.5.2; Bio-Rad) was used for densitometric analyses. Optical density values were normalized to total protein levels and considered as arbitrary units”.
The results part 2.2 starts analyzing signaling pathways; however, it is completely missing why the authors look at those pathways and what the single proteins are. An expert on the subject, of course, knows that but anyone else not.
Thank you for this point. According to the reviewer´s suggestion, the 2.2 section have been rewritten, including more extensive explanations in order to facilitate the understanding of the rationale as well as the significance of the obtained results (lines 167-196). Briefly, we monitor AKT (Protein kinase B) and ERK (extracellular-signal-regulated kinase 1/2) because they appear as main hubs in the interactome constituted by differentially expressed proteins detected by mass-spectrometry (Figure 3A). The other kinases tested are deregulated at the level of the olfactory bulb in other neurodegenerative phenotypes (Alzheimer´s and Parkinson´s diseases), so we have considered appropriate to analyse them in the context of Dementia with Lewy bodies. These kinase are: p38 MAPK (p38 mitogen-activated protein kinase), SEK1 (Mitogen-activated protein kinase Kinase 4), SAPK/JNK (stress-activated protein kinase/Jun-amino terminal kinase), PKA (Protein kinase A), MKK3 (Dual specificity mitogen-activated protein kinase kinase 3) and MKK6 (Dual specificity mitogen-activated protein kinase kinase 6).
To facilitate the reading of the new paragraphs, complete kinases names are detailed in the text and included in the abbreviations section of the new version of the manuscript.
Why is the manuscript organized in Results and Discussion together? In fact, a real discussion and implementation with the literature up to date is lacking.
As suggested by the reviewer, we have incorporate different paragraphs (section 2.2, section 2.3 and section 4) to increase the significance of the obtained results respect to the current knowledge in the field. As IJMS journal guidelines indicate that a combined Results and Discussion section is appropriate, we have decided to maintain the initial structure of the manuscript.
These references have been incorporated in the new version of the manuscript:
- IG McKeith, D Galasko, K Kosaka et al. Consensus guidelines for the clinical and pathologic diagnosis of dementia with Lewy bodies (DLB): report of the consortium on DLB international workshop. Neurology 1996 Nov;47(5):1113-24. doi: 10.1212/wnl.47.5.1113.
-Ian McKeith, Jacobo Mintzer, Dag Aarsland et al. Dementia with Lewy bodies. Lancet Neurol 2004 Jan;3(1):19-28. doi: 10.1016/s1474-4422(03)00619-7.
- IG McKeith, D W Dickson, J Lowe, M Emre et al. Diagnosis and management of dementia with Lewy bodies: third report of the DLB Consortium. Neurology 2005 Dec 27;65(12):1863-72. doi: 10.1212/01.wnl.0000187889.17253.b1. Epub 2005 Oct 19.
- Mary Catherine Mayo, Yvette Bordelon. Dementia with Lewy bodies. Semin Neurol. 2014 Apr;34(2):182-8. doi: 10.1055/s-0034-1381741. Epub 2014 Jun 25.
- Hiroshige Fujishiro, Shinichiro Nakamura, Kiyoshi Sato, Eizo Iseki. Prodromal dementia with Lewy bodies. Geriatr Gerontol Int 2015 Jul;15(7):817-26. doi: 10.1111/ggi.12466. Epub 2015 Feb 17.
- P C Donaghy, J T O'Brien, A J Thomas. Prodromal dementia with Lewy bodies. Psychol Med 2015 Jan;45(2):259-68. doi: 10.1017/S0033291714000816. Epub 2014 Apr 3.
- Thomas G Beach, Charles H Adler, Nan Zhang et al. Severe hyposmia distinguishes neuropathologically confirmed dementia with Lewy bodies from Alzheimer's disease dementia. PLoS One 2020 Apr 22;15(4):e0231720. doi: 10.1371/journal.pone.0231720. eCollection 2020.
- D C Davies, J W Brooks, D A Lewis. Axonal loss from the olfactory tracts in Alzheimer's disease
Neurobiol Aging Jul-Aug 1993;14(4):353-7. doi: 10.1016/0197-4580(93)90121-q.
- Philipp A Thomann, Vasco Dos Santos, Ulrich Seidl, Pablo Toro et al. MRI-derived atrophy of the olfactory bulb and tract in mild cognitive impairment and Alzheimer's disease. J Alzheimers Dis 2009;17(1):213-21. doi: 10.3233/JAD-2009-1036.
- Donna J Cross, Yoshimi Anzai, Eric C Petrie et al. Loss of olfactory tract integrity affects cortical metabolism in the brain and olfactory regions in aging and mild cognitive impairment. J Nucl Med
2013 Aug;54(8):1278-84. doi: 10.2967/jnumed.112.116558. Epub 2013 Jun 26.
-Praveen Bathini, Antoine Mottas, Muriel Jaquet, Emanuele Brai, Lavinia Alberi. Progressive signaling changes in the olfactory nerve of patients with Alzheimer's disease. Neurobiol Aging 2019 Apr;76:80-95. doi: 10.1016/j.neurobiolaging.2018.12.006. Epub 2018 Dec 27.
- Christoph Scherfler, Michael F Schocke et al. Voxel-wise analysis of diffusion weighted imaging reveals disruption of the olfactory tract in Parkinson's disease. Brain 2006 Feb;129(Pt 2):538-42. doi: 10.1093/brain/awh674. Epub 2005 Nov 4.
Sincerely yours,
Enrique Santamaría, Ph.D.
Clinical Neuroproteomics Unit, Navarrabiomed, Navarra Institute for Health Research (IdiSNA), Pamplona, Spain. E-mail: esantamma@navarra.es (Phone: +(34) 848 42 57 40)
Round 2
Reviewer 2 Report
The authors have properly replied to all the reviewers' concerns.